∂ | **Open Peer Review** | Environmental Microbiology | Research Article

# Rescue of morphological defects in *Streptomyces venezuelae* by the alkaline volatile compound trimethylamine

**Yanping Zhu,**[1] **Yanhong Zeng,**[1] **Meng Liu,**[2] **Ting Lu,**[1] **Xiuhua Pang**[1]

**ABSTRACT** Microorganisms can produce a vast diversity of volatile organic compounds of different chemical classes that are capable of mediating intra- and inter-kingdom interactions. In this study, we showed that the soil-dwelling bacterium *Streptomyces venezuelae* can produce alkaline volatiles under multiple growth conditions, which we discovered through investigation of the *S. venezuelae* mutant strain MU-1. Strain MU-1 has a defective morphology and exhibits a bald phenotype due to the lack of aerial mycelia and spores, as confirmed by scanning electron microscopy. Using physical barriers to separate the strains on culture plates, we determined that volatile compounds produced by wild-type *S. venezuelae* could rescue the phenotype of strain MU-1, and pH analysis of the growth medium indicated that these volatile compounds were alkaline. Ultra-high-performance liquid chromatography, combined with mass spectrometry analysis, showed that wild-type *S. venezuelae* produced abundant levels of the alkaline volatile trimethylamine (TMA) and the oxide form TMAO; however, the levels of these compounds were much lower in strain MU-1. Notably, exposure to TMA alone could rescue the phenotype of this mutant strain, restoring the production of aerial mycelia and spores. We also showed that the rescue effect by alkaline volatiles is mostly species-specific, suggesting that the volatiles may aid particular mutants or other less-fit variants of closely related species to resume normal physiological status and to compete more effectively in complex communities such as soil. Our study reveals a new and intriguing role for bacterial volatiles, including volatiles that may have toxic effects on other species.

**IMPORTANCE** Bacterial volatiles have a wide range of biological roles at intra- or inter-kingdom levels. The impact of volatiles has mainly been observed between producing bacteria and recipient bacteria, mostly of different species. In this study, we report that the wild-type, soil-dwelling bacterium *Streptomyces venezuelae*, which forms aerial hypha and spores as part of its normal developmental cycle, also produces the alkaline volatile compound trimethylamine (TMA) under multiple growth conditions. We showed that the environmental dispersion of TMA produced by *S. venezuelae* promotes the growth and differentiation of growth-deficient mutants of the same species or other slowly growing *Streptomyces* bacteria, and thus aids in their survival and their ability to compete in complex environmental communities such as soil. Our novel findings suggest a potentially profound biological role for volatile compounds in the growth and survival of communities of volatile-producing *Streptomyces* species.

**KEYWORDS** trimethylamine, alkaline volatile, *Streptomyces*, morphology

V olatile organic compounds are usually low-molecular-weight molecules, some of which have an unpleasant odor. Due to their low boiling points and high vapor pressure, these molecules can spread easily in soil, air, and water. Although thousands of volatiles have been identified, most volatiles can be categorized into a limited number

Address correspondence to Xiuhua Pang, pangxiuhua@sdu.edu.cn.

The authors declare no conflict of interest.

of chemical classes, including alcohols, aromatic compounds, ketones, terpenes, organic acids, esters, and others (1). Microorganisms, including bacteria and fungi, are major producers of volatiles (2, 3), and diverse biological roles have been reported for bacterial volatiles, which are manifested in communication, cooperation, and competition at intra- or inter-kingdom levels (2–4). However, the production of volatiles by microorganisms is very conditional, and the process can be influenced by a multitude of factors, such as temperature, pH, and medium components (5).

*Streptomyces* are soil-dwelling bacteria that are best known as a major source for bioactive compounds, including anti-infective agents and antibiotics, and for their ability to undergo cellular differentiation at late stages of development (6, 7). *Streptomyces* can also produce high amounts of volatile ammonium, which enables the long-range killing of a wide group of competing bacteria including Gram-positive and Gram-negative species (8). Surprisingly, *Streptomyces* volatiles were even found to initiate a previously unknown form of cellular development (9, 10). *Streptomyces* bacteria are generally considered to have three developmental stages in their life cycle, including vegetative hyphae, aerial hyphae, and spores. Development of vegetative hyphae and aerial hyphae are controlled by *bld* and *whi* genes, respectively (11). However, *S. venezuelae* can develop an "exploration stage" upon interaction with yeast (9); *Streptomyces* cells in this exploration stage can produce the alkaline volatile compound trimethylamine (TMA), and these explorer cells can communicate the exploratory behavior to other physically separated *Streptomyces* cells at other developmental stages using TMA (9). TMA exposure can also modify antibiotic resistance (12). However, it was demonstrated that TMA is produced by *S. venezuelae* only under very specific growth conditions, such as upon depletion of glucose in the medium (9, 13).

The impact of volatiles has mainly been observed between producing bacteria and recipient (perceiving) bacteria of different species (2, 14). In this study, we demonstrated that *S. venezuelae* can produce the alkaline volatile TMA under multiple growth conditions, with or without glucose, and we further found that the TMA produced by *S. venezuelae* can rescue the morphological defects of a mutant strain of the same species, revealing a new role for bacterial volatiles.

## RESULTS

### *S. venezuelae* mutant strain MU-1 has a defective morphology

As part of our ongoing genetics studies on *S. venezuelae* genes (15), we generated a mutant strain of *sven*_1780-81 in *S. venezuelae* strain ISP5230 through homologous recombination following standard protocols as described for mutation of other genes in *Streptomyces* (16); however, the mutation could not be complemented, which suggested the presence of at least one secondary mutation in the resulting *S. venezuelae* mutant, hereafter designated as MU-1. The confluent growth of the parental *S. venezuelae* strain ISP5230 (WT) culture on solid N-Evans-CA medium appeared fluffy and greenish due to the formation of the aerial mycelium and mature spores (Fig. 1A; Fig. S1A). In contrast, strain MU-1 appeared bald even after prolonged incubation under the same growth conditions (Fig. 1A; Fig. S1B), a phenotype very similar to that of *bld* strains, indicating that strain MU-1 had morphological defects in the formation of the aerial mycelium and spores.

To verify the morphological defect of MU-1, we examined the morphology of MU-1 by scanning electron microscopy (SEM) using strains grown for 4 days on solid N-Evans-CA medium. The WT *S. venezuelae* strain was abundantly covered with the typical straight spore chains with visible segmentation (Fig. 1B). However, aerial mycelium was minimal in MU-1, and vegetative mycelium, which is distinct from aerial mycelium with spore chains, predominated (Fig. 1B), confirming that MU-1 is defective in the formation of aerial mycelia and spores, explaining the bald phenotype of this strain.

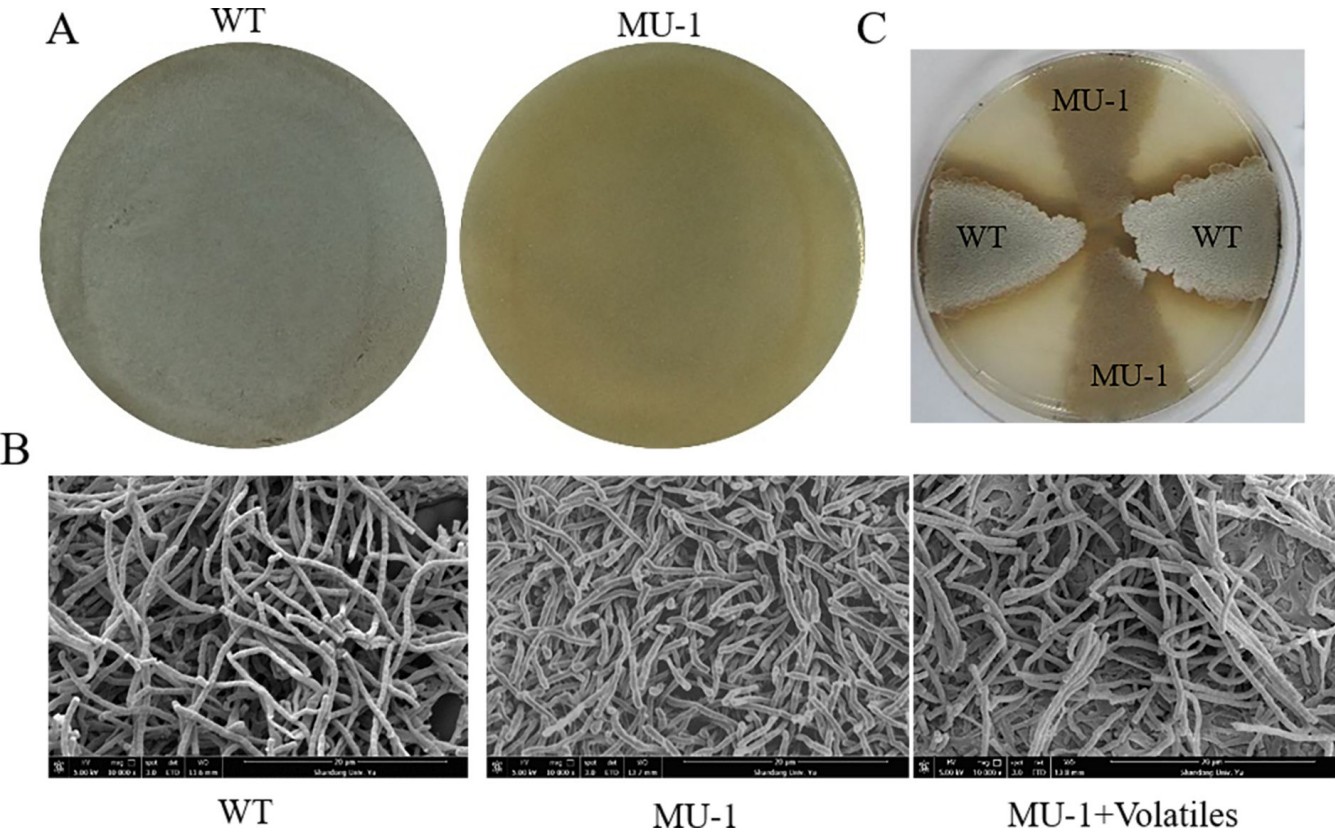

**FIG 1** The *S. venezuelae* mutant strain MU-1 has a defective morphology. (A) Phenotypes of the *S. venezuelae* wild-type strain ISP5230 (WT) and the mutant strain MU-1 grown at 30°C on solid N-Evans-CA medium (120 h) on separate plates. (B) SEM images of ISP5230 (WT) and MU-1 after growth on N-Evans-CA agar for 5 days, and MU-1 after 10 days of co-incubation with ISP5230 on N-Evans-CA agar (MU-1 + volatiles). (C) Phenotypes of alternating patches of strain ISP5230 (WT) and MU-1 grown at 30°C on the same plate of solid N-Evans-CA medium (96 h).

## Co-incubation with the WT *S. venezuelae* strain rescues the defective morphology of MU-1

Although strain MU-1 appeared bald after prolonged growth when cultured separately from the WT strain (Fig. 1A), we noticed that, when streaked on the same plate with the WT strain, some edges of the MU-1 patches that directly contacted the WT patch turned greyish after a short period of growth (Fig. 1C), indicating that the cells in these edge regions underwent further development and formed aerial mycelium and spores, which suggested that the WT *S. venezuelae* strain could promote the growth of MU-1. To further investigate this observation, we inoculated the WT and MU-1 strains onto a standard petri dish, with each strain occupying half of the plate (Fig. 2A; Fig. S2A). The WT patch turned greenish (indicating formation of mature spores) by 72 h of growth, while the MU-1 patch was still bald at that time (Fig. 2A; Fig. S2A). However, the edge of the MU-1 culture closest to the WT patch started to turn grey (indicating formation of aerial hyphae) by 96 h of co-incubation, and nearly all of the MU-1 patch turned a greenish color by 144 h and later time points (Fig. 2A; Fig. S2A), indicating that co-incubation with the WT strain can restore the developmental phenotype of the mutant MU-1 to a level comparable to that of the WT strain.

We hypothesized that the rescue of the morphological defects of MU-1 could be caused either by the diffusion of soluble materials secreted by the WT strain, by the spread of volatile compounds produced by the WT strain, or by both mechanisms. To determine the underlying mechanism(s), we inoculated the WT and MU-1 strains on a petri dish with a physical barrier that can block the diffusion of soluble materials but allows the spread of volatile molecules (Fig. 2B; Fig. S2B). Strain MU-1 started turning

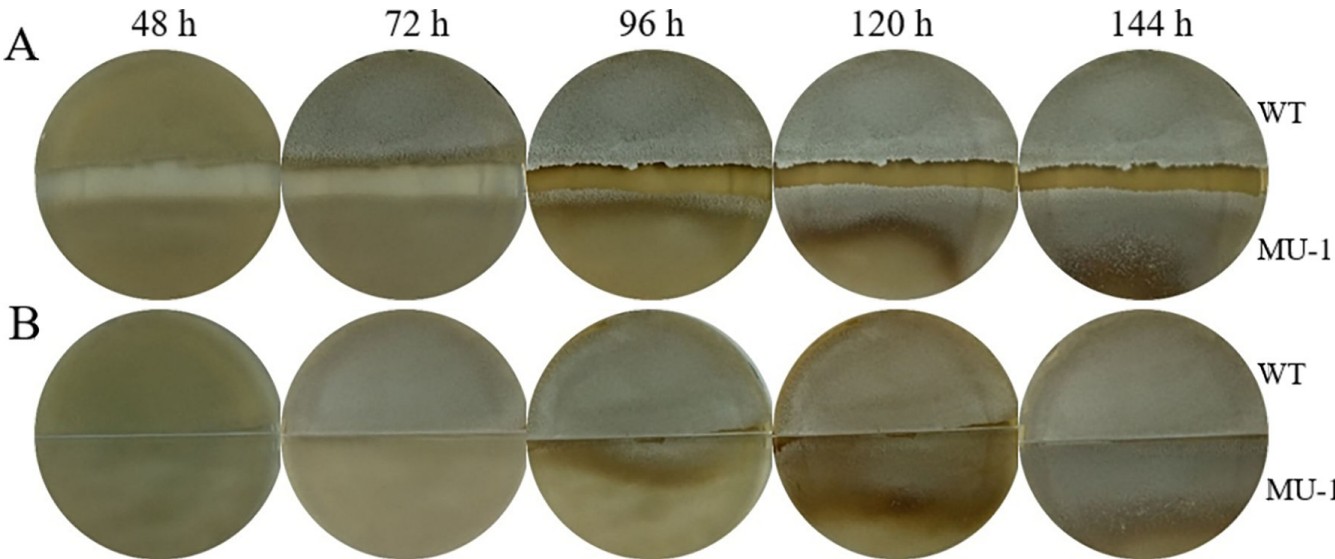

**FIG 2** Co-incubation with *S. venezuelae* strain ISP5230 rescues the morphological defects of MU-1. (A, B) Strain ISP5230 (WT) and MU-1 were grown at 30°C on solid N-Evans-CA medium on (A) a standard petri dish without a physical barrier and (B) a special petri dish with a physical barrier between the two strains. Images of the top of the plate were taken at the indicated times.

greenish, an indicator of aerial mycelium and spore formation and similar to the pattern observed on the standard petri dish (Fig. 2A); by 144 h and longer times of co-incubation, the MU-1 section of the plate closely resembled that of WT (Fig. 2B; Fig. S2B), suggesting that air-borne volatile compounds produced by the WT strain had rescued the morphological defects of MU-1.

We also examined this rescuing effect by SEM scanning (Fig. 1B), which showed that the rescued MU-1 had formed aerial mycelium and straight spore chains like those of the WT strain (Fig. 1B), confirming that co-incubation with the WT strain restored a normal growth phenotype to MU-1.

## Production of alkaline volatile compounds by *S. venezuelae*

*Streptomyces* species are known to be a rich source for volatile compounds (8). To determine whether the volatiles produced by *S. venezuelae* under the tested conditions were acidic or alkaline, we inoculated the WT strain onto one half of a petri dish with a physical barrier, leaving only blank (uninoculated) medium on the other half, and the pH value of the medium on both halves was measured at different growth times (Fig. 3A). The growth medium of the WT strain displayed a steady increase in pH value from an initial neutral pH of about pH 6.9 at 48 h to the peak pH value of about pH 8.6 at 168 h, which was then followed by a relatively steady pH level afterwards (Fig. 3A), suggesting that the *S. venezuelae* WT strain produced basic (high pH) compounds during growth. The pH value of the blank medium also increased steadily, but at a much slower rate than with the WT strain, only reaching a pH value comparable to that of the section of the plate with the WT strain after prolonged incubation (Fig. 3A). Altogether, these data indicated that the *S. venezuelae* WT strain produced alkaline volatile compounds and that absorption of these volatile compounds increased the pH of the uninoculated medium on the same plate.

## TMA exposure rescues the morphological defects of the mutant strain MU-1

It has been reported that multiple *Streptomyces* strains, including *S. venezuelae*, are able to produce ammonium (8). To test if ammonium was responsible for rescuing the growth phenotype of MU-1, we exposed a plate seeded with MU-1 to ammonium, using water as a control, to evaluate the effects on MU-1 growth (Fig. 4). The morphological defects of

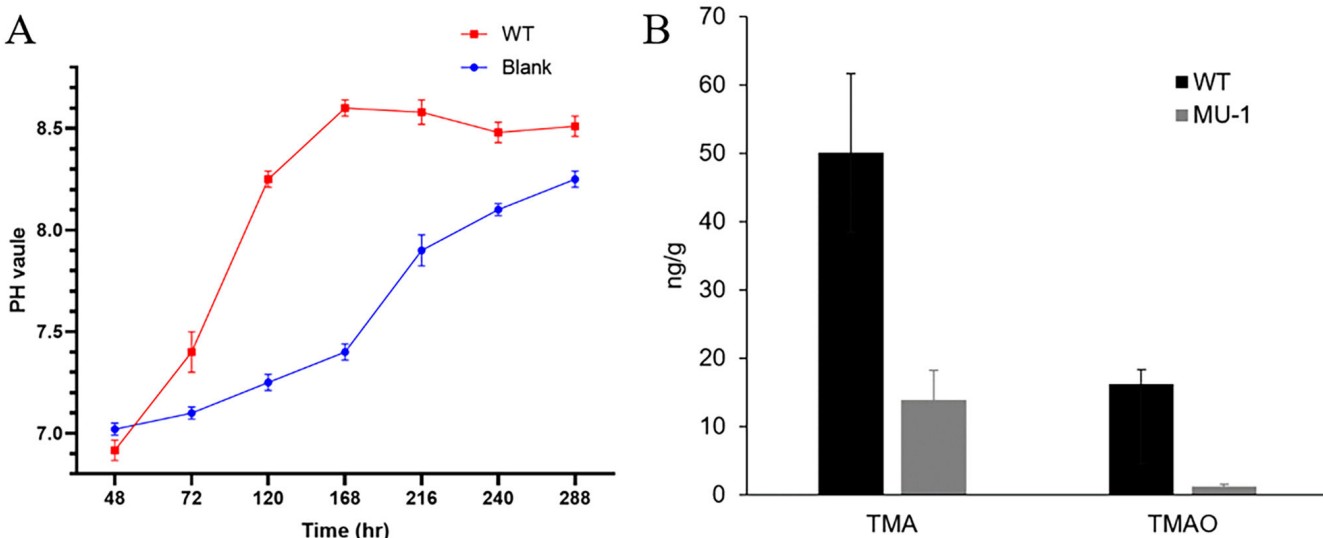

**FIG 3** The *S. venezuelae* WT strain ISP5230 produces TMA (trimethylamine). (A) The temporal pH values of the *S. venezuelae* strain ISP5230 (WT) growth medium and the blank (uninoculated) medium. (B) Quantification of TMA production. *S. venezuelae* strains ISP5230 (WT) and MU-1 were grown at 30°C on solid N-Evans-CA medium for 10 days. The mycelia were harvested, and TMA or its oxide form (TMAO) was quantified by UHPLC. Data are the means with standard deviations from six different preparations.

MU-1 were not altered even after prolonged exposure to ammonium (Fig. 4A; Fig. S3A), and it appeared that the growth of MU-1 was even inhibited to some degree, compared with the control sample exposed to water (Fig. 4B; Fig. S3B), indicating that ammonium is not the volatile that rescues the growth defects of MU-1. Our analysis indicated that the growth medium of *S. venezuelae* is alkaline under our tested conditions (Fig. 3A), and in addition to ammonium, *S. venezuelae* is known to produce another alkaline compound, TMA, although under specific growth conditions, that is, on YP(G-) medium depleted of glucose and when triggered by interaction with yeast (9, 13). We therefore also tested the potential of TMA to rescue the MU-1 defects under the same conditions (Fig. 4C; Fig. S3C). In contrast to the sample exposed to ammonium, the cultures of MU-1 exposed to TMA appeared greyish by 84 h (Fig. S3C), indicating formation of aerial mycelium, although in a very limited area. Additionally, most of the culture of MU-1 growing closest to the TMA container turned greenish after prolonged incubation (Fig. 4C; Fig. S3C), indicating that TMA exposure promoted the growth of MU-1.

We also measured the pH value of the MU-1 growth medium following exposure to ammonium or TMA. Ammonium and TMA raised the pH value of the MU-1 growth medium similarly (Fig. S4), and the MU-1 growth medium was already over pH 8 after 72 h of exposure to ammonium and TMA (Fig. S4). The media in the three MU-1 plates exposed for 10 days to ammonium averaged pH 8.63, which is close to the pH value of the MU-1 growth medium after exposure to TMA for 10 days (average pH 8.85); both pH averages were comparable to the pH of the WT culture at 10 days of growth (Fig. 3A). In contrast, the medium with the control samples averaged pH 4.49 after 10 days. These data suggested that the pH itself was not the cause of the growth promotion of MU-1 since TMA promoted growth whereas ammonium did not. In conclusion, our data suggested that the WT *S. venezuelae* strain produces TMA under our tested conditions, and it is TMA, rather than ammonium or a basic pH, that induces the morphological changes in strain MU-1, rescuing the growth phenotype.

## WT *S. venezuelae* produces TMA under multiple growth conditions

Both TMA exposure alone and the volatiles produced by the *S. venezuelae* WT strain were able to rescue the defective MU-1 phenotype; however, our growth medium includes

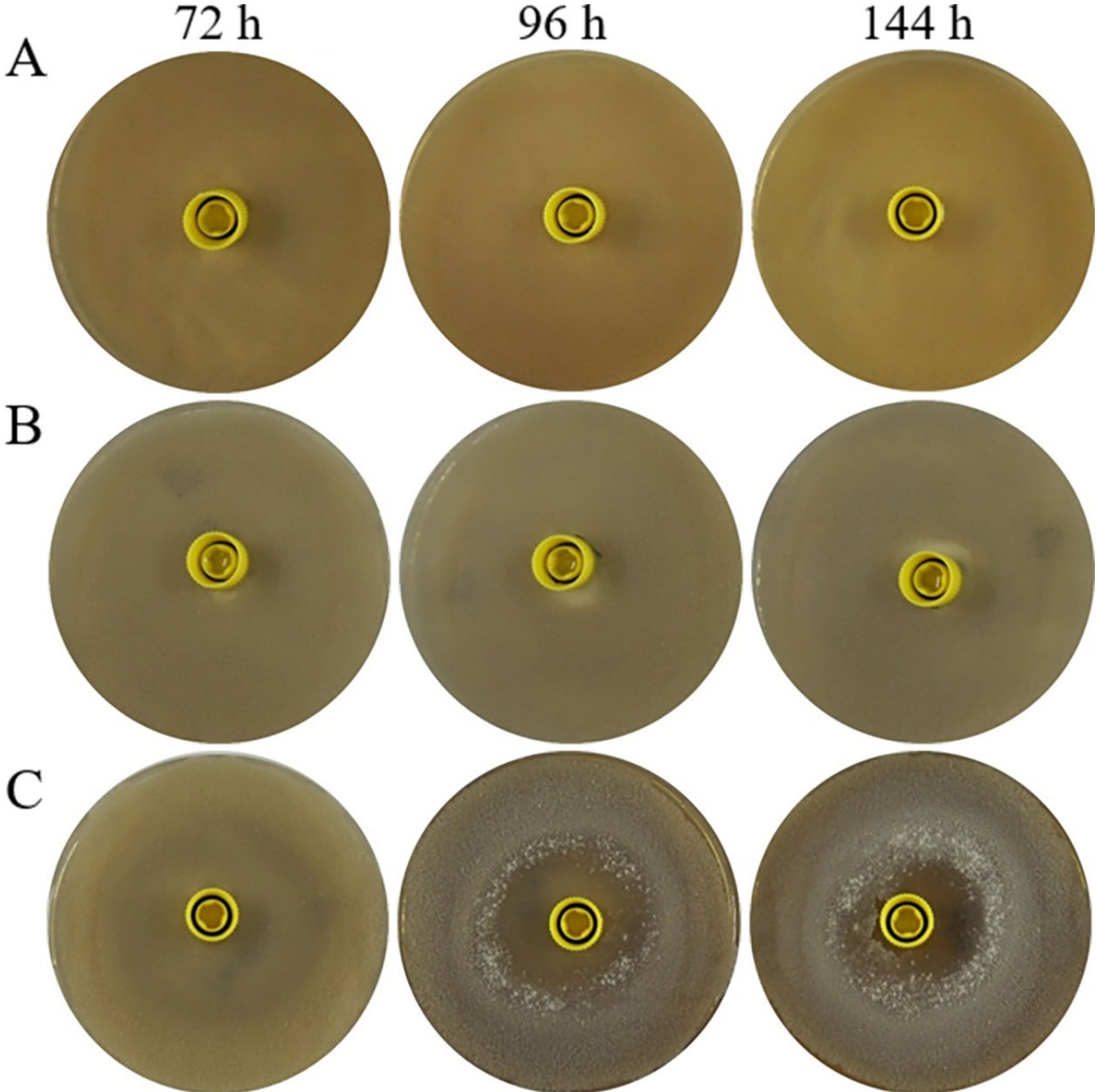

**FIG 4** The effects of exposure to ammonium (A), water (B), and TMA (C) on the growth of strain MU-1 at 30°C on solid N-Evans-CA medium. Water (control) or the tested reagents were added to the yellow container in the center of the plate, and plates were incubated for the indicated times.

glucose and therefore differs from the glucose-depleted conditions that were reported to be required for TMA production (9). Consequently, we were curious whether the *S. venezuelae* WT strain produces TMA on our growth medium, which was N-Evans-CA. We performed targeted, absolute quantification of TMA using *S. venezuelae* WT and MU-1 strains grown on N-Evans-CA. Our data showed that the level of TMA produced by the WT strain was 3.6-fold higher than that of MU-1 (wet cell weight), and the level of TMAO, the oxide form of TMA, for WT was 13.3-fold higher than for MU-1 (Fig. 3B), indicating that the WT strain produces abundant TMA on our growth medium, which includes glucose.

Production of volatiles by *Streptomyces* is reportedly conditional (8, 9). Therefore, in addition to N-Evans-CA, we tested the rescue of the MU-1 morphological defects by the WT strain grown on multiple types of media routinely used for *Streptomyces* growth (16, 17). For each plate, since MU-1 exhibits the bald phenotype on N-Evans-CA agar, this medium was used on half of the plate with MU-1, and the medium was varied on the other half for the WT strain. Co-incubation with the WT strain rescued the defects of MU-1 when grown on YBP (Fig. 5A; Fig. S5), a rich growth medium with added glucose (17), suggesting that the WT can produce TMA during growth on solid YBP medium. Similarly, when using MS medium, a complex medium for *Streptomyces* growth that lacks glucose (16), the morphological defects of MU-1 were also rescued (Fig. 5B; Fig. S6), suggesting that the WT produces TMA when grown on solid MS medium. However, during co-incubation with the WT strain on MYM, a medium often used for growing *S. venezuelae* and that does not have added glucose (18), the defects of MU-1 were not

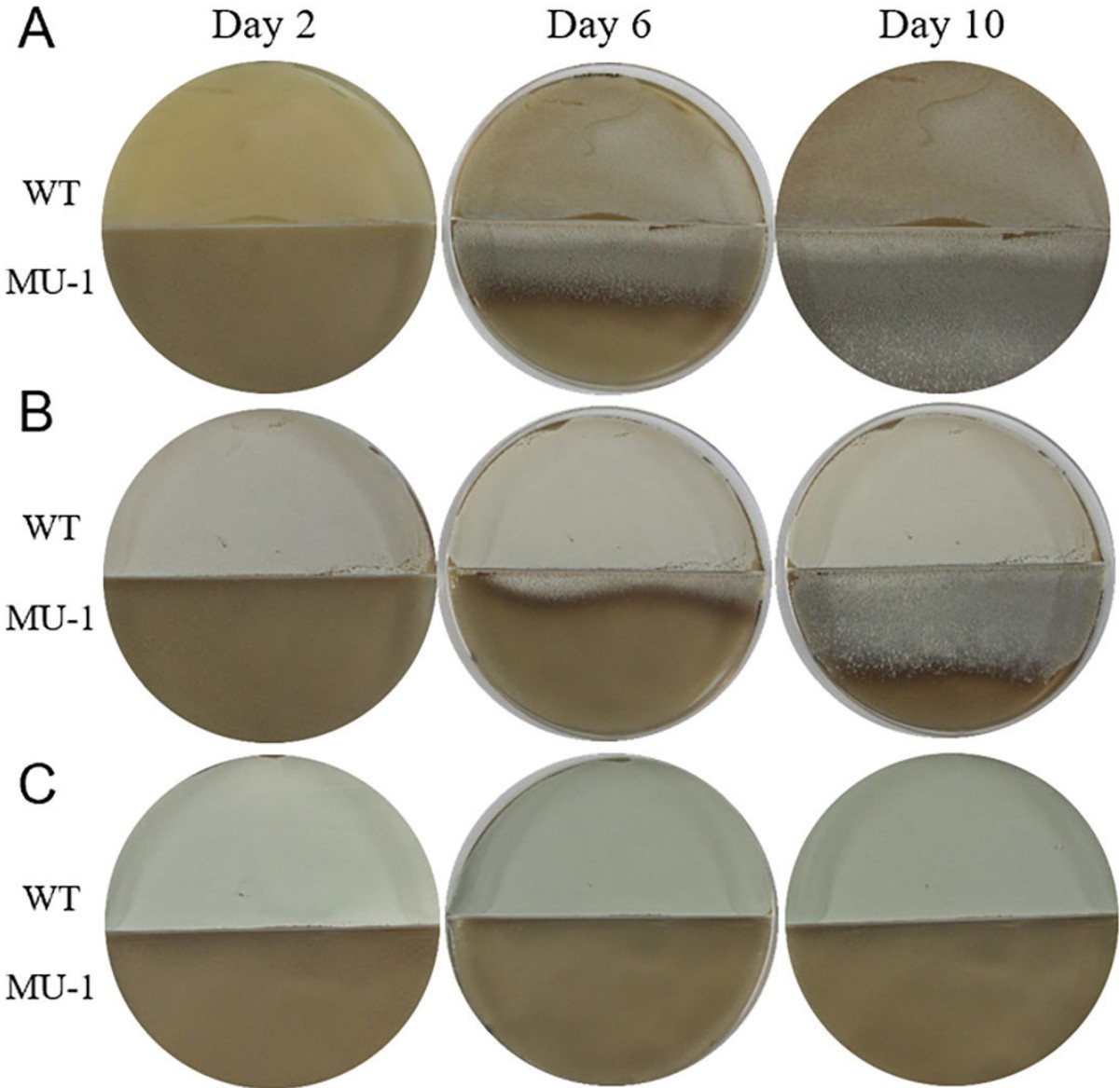

**FIG 5** Evaluation of the rescue of the morphological defects of MU-1 by *S. venezuelae* strain ISP5230 on different media. Co-incubation of *S. venezuelae* strain ISP5230 (WT) grown on YBP (A), MS (B), and MYM (C) media with strain MU-1 grown on N-Evans-CA medium. The petri dishes contained physical barriers between the two strains.

rescued even after prolonged incubation time (Fig. 5C; Fig. S7), suggesting that the *S. venezuelae* WT strain does not produce TMA on this medium or that the level of TMA is below the threshold needed to induce the morphological changes in MU-1. The pH values of the various types of agar media with *S. venezuelae* WT strain were measured after 10 days of growth, averaging pH 8.58 on YBP and pH 8.77 on MS, consistent with the production of the alkaline compound TMA under these two conditions. However, the average on MYM was only pH 6.98, indicating that no alkaline compound was produced or accumulated, which is in agreement with the failure of the WT strain to rescue the morphology of MU-1 when grown on MYM. Overall, our data showed that the *S. venezuelae* WT strain can produce TMA on multiple, although not all, types of routinely used media for growth of *Streptomyces* and that the presence of glucose is not essential under the tested conditions.

## Co-incubation does not rescue all *Streptomyces* strains with morphological defects

Given the ability of the alkaline volatile TMA produced by WT *S. venezuelae* strain to rescue the morphological defects of MU-1, we wanted to know if the same effect would occur with other *Streptomyces* mutants that have similar morphological defects. To address this issue, we co-incubated the *S. venezuelae* WT strain with four mutant *Streptomyces* strains that have a bald phenotype: $\Delta mtrA_{SVEN}$, derived from *S. venezuelae* ISP5320, has a deletion of the gene encoding the response regulator MtrA, which functions as a critical developmental regulator (19); $\Delta 5003_{SVEN}$, also derived from *S. venezuelae* ISP5320, has a deletion in the gene for the newly identified major developmental regulator SVEN_5003 (15); and the *S. coelicolor* mutant strains $\Delta mtrA_{SCO}$ and $\Delta orrA_{SCO}$, both derived from *S. coelicolor* M145, have a deletion in the response regulator gene *mtrA* and the gene encoding the orphan response regulator OrrA (SCO3008), respectively (19, 20). If these *S. venezuelae* or *S. coelicolor* mutants were rescued by co-incubation, we would expect to see a grey to greenish color indicative of aerial mycelium and spore formation, as found for MU-1. However, our co-incubation tests showed that the phenotype of none of these strains was altered even after prolonged co-incubation with the *S. venezuelae* WT strain (Fig. 6; Fig. S8 to S11), implying that the rescue of MU-1 by the WT strain via TMA was strain-specific. We also measured the pH values of the cultures of these four mutants after growth on YBP for 10 days. Values averaged pH 8.53 for $\Delta mtrA_{SVEN}$, pH 8.46 for $\Delta 5003_{SVEN}$, pH 8.64 for $\Delta mtrA_{SCO}$, and pH 8.45 for $\Delta orrA_{SCO}$, indicating absorption of alkaline TMA in their growth media, although the normal morphology was not restored. Collectively, our data showed that the rescue of the morphological defects of MU-1 via the alkaline volatiles produced by *S. venezuelae* is strain-specific.

## Co-incubation of MU-1 with other *Streptomyces* species

To determine the potential of other *Streptomyces* species, in addition to *S. venezuelae*, to rescue the defects of MU-1 through the production of TMA, we co-incubated MU-1 with multiple other *Streptomyces* species, including *S. coelicolor* M145, *S. lividans* 1326, *S. avermitilis*, *S. hygroscopicus* 5008, and *Streptomyces* sp. FR008, and observed the phenotype of MU-1. However, none of these *Streptomyces* strains altered the phenotype of MU-1 under the tested conditions, even after prolonged incubation time (Fig. S12 to S16), suggesting that either these *Streptomyces* species do not produce TMA or the levels of TMA they produce are not high enough to promote the growth of MU-1.

## Co-incubation of *S. venezuelae* with other *Streptomyces* species

Given the dramatic phenotype change in MU-1 induced by the alkaline volatiles produced by the WT *S. venezuelae* strain, we wondered if these alkaline volatiles would impact the growth of other *Streptomyces* species that did not produce TMA, as determined by their inability to rescue MU-1 in the preceding co-incubation experiment.

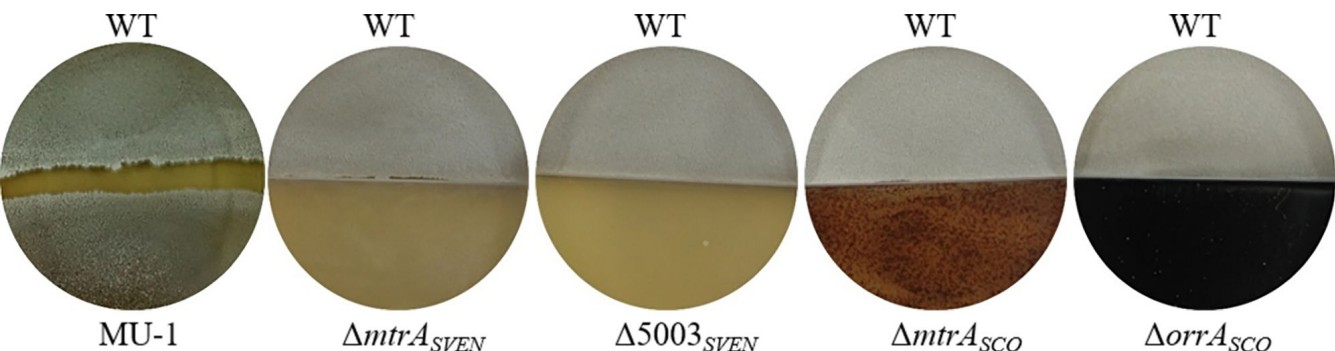

**FIG 6** Evaluation of the rescue of different strains with a bald phenotype by *S. venezuelae* Strain ISP5230. Strain ISP5230 (WT) and MU-1 were grown on solid N-Evans-CA medium for 6 days. The two *S. venezuelae* mutant strains ($\Delta mtrA_{SVEN}$ and $\Delta 5003_{SVEN}$) and the two *S. coelicolor* mutant strains ($\Delta mtrA_{SCO}$ and $\Delta orrA_{SCO}$) were grown on YBP medium and incubated with strain ISP5230 in petri dishes with physical barriers for 10 days.

Therefore, we grew *S. venezuelae* strain ISP5230 on MS, a medium on which ISP5230 produces TMA, and co-incubated *S. coelicolor*, *S. lividans*, *S. hygroscopicus*, and *Streptomyces* sp. FR008 with the ISP5230 cultures on the barrier-containing plates. We then monitored these strains for potential changes. No obvious change was observed for *S. coelicolor* M145, *S. hygroscopicus* 5008, or *Streptomyces* sp. FR008 even after prolonged co-incubation with strain ISP5230 (Fig. S17 to S19), implying that the alkaline volatiles produced by *S. venezuelae* do not elicit phenotypic changes in these *Streptomyces* species or that the changes are too minor to be visible under the tested conditions. However, we also tested *S. lividans* TK24. Strain TK24 started to form aerial mycelium (visible as white material on the plate) after 4 days of co-incubation with ISP5230 (Fig. 7; Fig. S20). In contrast, no change was observed for TK24 grown alone even after 10 days of incubation, with only faint white haze observed throughout the incubation period (Fig. 7; Fig. S20). This finding suggests that the alkaline volatiles produced by ISP5230 can promote the growth of some other *Streptomyces* species.

## DISCUSSION

A wide range of biological roles have been reported for bacterial volatiles (2–4), and most often, the effect of volatiles is apparently uni-directional at the intra- and cross-kingdom levels, reflected by changes in the organisms perceiving the volatiles rather than the producer itself (21). In this study, we demonstrated that *Streptomyces* volatiles can rescue the morphological defects of a mutant strain of the same species as the producer, a new role not previously observed for bacterial volatiles. We also showed that the rescue is dependent on the medium used for the growth of *S. venezuelae*, indicating that production of the alkaline volatiles is conditional, a phenomenon often observed for production of other bacterial volatiles (8). We further showed that the alkaline volatiles produced by *Streptomyces* include TMA and that TMA played a major role in rescuing the defects of MU-1. Although the capacity of *S. venezuelae* to produce TMA was previously reported, in those studies, TMA was produced under glucose-depleted conditions and while interacting with yeast (9, 13). However, we showed that TMA can be produced by *S. venezuelae* under conditions with or without glucose, expanding the range of TMA production conditions, and hence suggesting a broader potential function for TMA in bacterial physiology.

Due to their large genome sizes (22, 23), *Streptomyces* species are a huge reservoir for secondary metabolites, some of which are volatile compounds, including geosmin, which has a well-known earthy odor (24), although its biological role for the producer bacterium is not yet clear (25). Recently, it was found that the odor of geosmin, together with that of another volatile compound, 2-methylisoborneol, may help attract soil arthropods and thus increase the spread of the *Streptomyces* producer (26). The volatiles produced by a specific *Streptomyces* species can be a mixture of many different chemical molecules (9, 22). Although we showed that TMA alone can restore the phenotype of

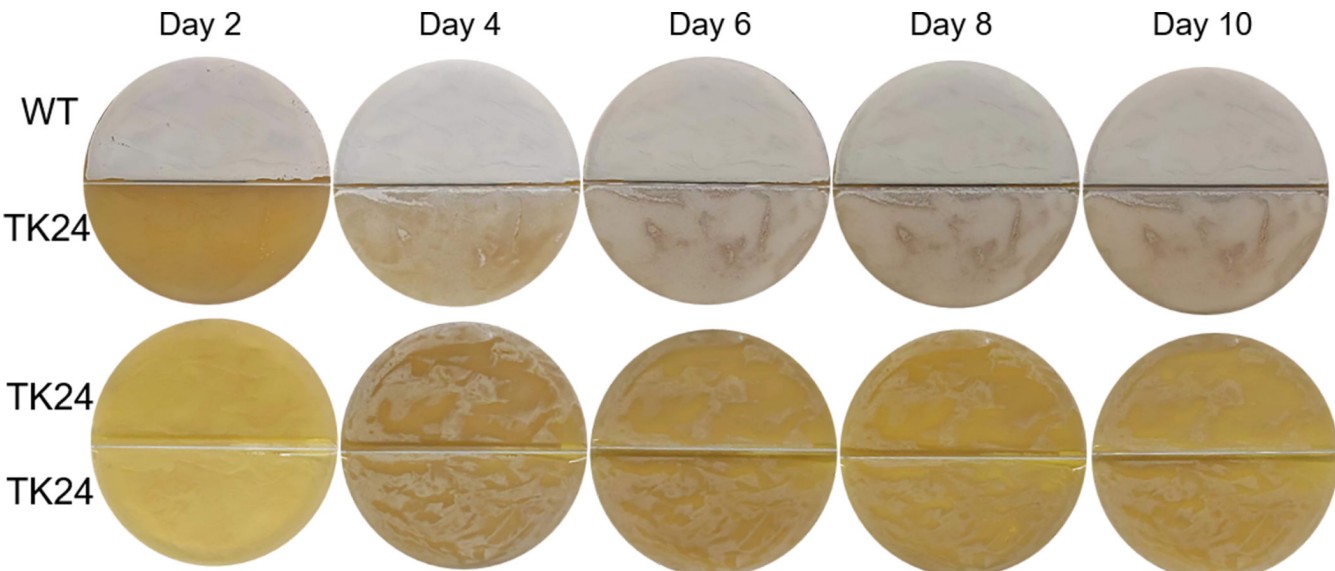

**FIG 7** Co-incubation with *S. venezuelae* strain ISP5230 promotes the growth of other *Streptomyces* species. Strain ISP5230 (WT) was grown on MS and *S. lividans* TK24 was grown on YBP at 30°C on a special petri dish with a physical barrier between the two strains. Images of the top of the plate were taken at the indicated times.

MU-1, but certainly other volatile compounds may contribute. However, identification of these other compounds requires further study.

In addition to the *S. venezuelae* mutant MU-1, exposure to the WT *S. venezuelae* strain induced the morphological development of the *S. lividans* strain TK24. *Streptomyces* may, therefore, produce volatile compounds to aid the development of bacteria of the same species or some close species that are at a growth disadvantage due to mutation or environmental conditions, which may, in turn, enable all of the species involved to compete more effectively against unrelated species in complex environments.

Due to the small size and relatively simple structure of volatile molecules, only a small number of genes would likely be required for their synthesis, making the identification of the associated genes more difficult. In contrast, genes involved in the biosynthesis of antibiotics, including those responsible for synthesizing the structure and for regulation, are usually clustered together, with these clusters ranging from tens of kb to more than 100 kb in length, making these clusters easier to identify and investigate (6). The genes for TMA synthesis have not yet been identified in *Streptomyces*. However, our study provides a helpful starting point by revealing strains and conditions that do or do not generate TMA and that could therefore be compared using transcriptomics and other analyses. Identifying the genes involved in TMA synthesis will be the focus of our future study.

## MATERIALS AND METHODS

### Culture conditions

*Streptomyces* strains were grown on several types of agar: N-Evans-CA agar, comprising Basic Evans (25 mM TES, 10 mM KCl, 2 mM NaSO$_4$, 2 mM citric acid, 0.25 mM CaCl$_2$, 1.25 mM MgCl$_2$, 1 mM NaMoO$_4$, and 0.5% Evans trace elements, pH 7.2) supplemented with glucose (2.5%) as carbon source, casamino acids (1%) as nitrogen source, and NaH$_2$PO$_4$ (2 mM) as phosphate source (27); YBP agar (2 g yeast extract, 2 g beef extract, 4 g Bacto peptone, 1 g MgSO4, 15 g NaCl, 10 g glucose, and 20 g agar in 1 L H$_2$O, pH 7.0) (17); MYM (maltose-yeast extract-malt extract) agar (10 g malt extract, 4 g yeast extract,

4 g maltose, and 20 g agar in 1 L $H_2O$, pH 7.4); and MS agar (20 g mannitol, 20 g soya flour, and 20 g agar in l L tap water, pH 7.0) (16).

## Co-incubation analysis and measurement of pH

The specified agar medium was poured into one half or both halves of petri dishes (Fig. S21), as indicated, with a polystyrene physical barrier across the diameter that allowed for some headspace between the top of the barrier and the lid. Equal numbers of *Streptomyces* spores were inoculated onto the agar and were incubated at 30°C at different times. To measure the pH of the solid agar following culture growth, the agar was cut into pieces, mixed with an equal amount of water, followed by vortexing for 2 h. The pH of the supernatant was then measured.

## Volatile-exposure assay

A total of $2 \times 10^6$ spores of mutant strain MU-1 were added to 300 µL of water, and the spores were then plated on N-Evans-CA agar using a cell spreader. After drying of the plates, a small cylindrical container was positioned in the center of each plate, and 200 µL of water, ammonium (1.2 M), or TMA (1.0 M) was added to the container. The plates with the filled containers were then incubated at 30°C, and images were taken after exposure to TMA, ammonium, or water at the indicated times. The containers were refilled every other day as needed to ensure that reagent was present.

## Quantification of TMA

Cultures of the WT strain and MU-1 were first grown on MS medium, on which the mutant strain MU-1 can produce spores. Equal numbers of spores ($2 \times 10^6$) of the wild-type strain *S. venezuelae* ISP5230 and MU-1 were then inoculated onto N-Evans-CA medium (six regular petri dishes for each strain) and were incubated at 30°C for 10 days before the mycelium of each strain was harvested and frozen at −80°C. Extraction and analysis of metabolites were then performed by the Shanghai Applied Protein Technology Co., Ltd. To extract the metabolites, 100 mg samples were weighed and placed in 2 mL centrifuge tubes after being thawed on ice. A 10 µL internal standard solution and 500 µL precooled acetonitrile-water (9/1, vol/vol) solution were added. Then steel balls were added to the mixture, and the mixture was homogenized two times (20 s each time), vortexed and mixed for 30 s; the homogenate was incubated at 4°C for 10 min. After protein precipitation, the homogenate was centrifuged at 14,000 RCF for 20 min at 4°C, and the supernatant was used for high-performance liquid chromatography–tandem mass spectrometry (HPLC–MS/MS) analysis.

The separation was performed on an ultra-high-performance liquid chromatography (UHPLC) system (Agilent 1290 Infinity UHPLC) on a HILIC column (Waters, BEH HILIC 2.5 µm, 2.1 mm × 100 mm column) by gradient elution. Eluent A was acetonitrile, and Eluent B was 10 mM ammonium formate buffer prepared in water (pH 3.5). The gradient elution program was as follows: 0 min = 90% B, 1.5 min = 90% B, 4.5 min = 87% B, 7 min = 85% B, 7.5 min = 50% B, 10 min = 50% B, 10.5 min = 90% B, and 14 min = 90% B. Before injecting the next sample, the column was equilibrated with the initial mobile phase for 5 min. The flow rate was constant at 0.4 mL/min, and the column temperature was set at 25°C. A 5500 QTRAP (AB SCIEX) was used in positive switch mode. The electrospray ionization source conditions were as follows: source temperature, 550°C; ion Source Gas1 (Gas1), 55; Ion Source Gas2 (Gas2), 55; Curtain gas, 40; ionSapary Voltage Floating, +4,500 V. The multiple reaction monitoring method was used for mass spectrometry quantitative data acquisition. MultiQuant or Analyst was used for quantitative data processing. The integration was further checked manually.

## Scanning electron microscopy

For SEM analysis, sterile coverslips were inserted into N-Evans-CA medium inoculated with spores, followed by incubation for 96 h for *S. venezuelae* ISP5230 or MU-1, and by incubation for 168 h for MU-1 co-incubated with ISP5230. Next, the coverslips were transferred to a solution containing 2.5% glutaric acid for fixation, followed by washing two to three times with PBS buffer (pH 7.0). Afterward, the coverslips were treated with PBS buffer containing 1% osmic acid for 2 h, followed by another wash with PBS buffer. A gradient of ethanol solution (30%, 50%, 70%, 80%, 90%, and 100%) was used for dehydrating the coverslips. Finally, the coverslips were dried by a critical point dryer (LEICA EM CPD300), coated with metal ions using ion sputtering and metal coating equipment (Cressinton Sputter Coater 108), and finally viewed with a scanning electron microscope (FEI Quanta 250 FEG, USA) as described (28).

## ACKNOWLEDGMENTS

This work was supported by grants from the Open Funding Project of State Key Laboratory of Microbial Metabolism (MMLKF24-04 to X.P.), from the China Postdoctoral Science Foundation (2023M732083 to Y. Zhu), from the National Natural Science Foundation of China (NSF32270072 to X.P.), and from Qingdao Postdoctoral Project Funding (QDBSH20230102103 to Y. Zhu).

## AUTHOR AFFILIATIONS

[1]The State Key Laboratory of Microbial Technology, Shandong University, Qingdao, China
[2]School of Municipal and Environmental Engineering, Shandong Jianzhu University, Jinan, China

## AUTHOR ORCIDs

Xiuhua Pang ⓘ http://orcid.org/0000-0002-0115-1973

## AUTHOR CONTRIBUTIONS

Yanping Zhu, Data curation, Funding acquisition, Investigation, Methodology | Yanhong Zeng, Investigation | Meng Liu, Investigation | Ting Lu, Investigation | Xiuhua Pang, Conceptualization, Formal analysis, Funding acquisition, Supervision, Writing – original draft, Writing – review and editing

## ADDITIONAL FILES

The following material is available online.

### Supplemental Material

**Supplemental material (Spectrum01195-24-s0001.pdf).** Fig. S1 to Fig. S21.

### Open Peer Review

**PEER REVIEW HISTORY (review-history.pdf).** An accounting of the reviewer comments and feedback.

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
