## [Reviewer comments · Microbiology Spectrum]

Microbiology Spectrum

Rescue of morphological defects in *Streptomyces venezuelae* by the alkaline volatile compound trimethylamine

Yanping Zhu, Yanhong Zeng, Meng Liu, Ting Lu, and Xiuhua Pang

Corresponding Author(s): Xiuhua Pang, Shandong University - Qingdao Campus

Review Timeline:

Submission Date:	May 13, 2024
Editorial Decision:	June 27, 2024
Revision Received:	July 17, 2024
Accepted:	July 24, 2024

Editor: Katharina Kujala

Reviewer(s): Disclosure of reviewer identity is with reference to reviewer comments included in decision letter(s). The following individuals involved in review of your submission have agreed to reveal their identity: Ban Al-Joubori (Reviewer #1)

Transaction Report:

DOI: <https://doi.org/10.1128/spectrum.01195-24>

Re: Spectrum01195-24 (Rescue of morphological defects in *Streptomyces venezuelae* by the alkaline volatile compound trimethylamine)

Dear Dr. Xiuhua Pang:

Thank you for the privilege of reviewing your work. Below you will find my comments, instructions from the Spectrum editorial office, and the reviewer comments.

I have received two reviews for your manuscript. As you will see, both reviewers were positive about your manuscript, but have raised some concerns and made suggestions for changes. If you feel that you can address these concerns, please resubmit an updated version of your manuscript following the guidelines below.

Revision Guidelines

Sincerely,
Katharina Kujala
Editor
Microbiology Spectrum

Reviewer #1 (Comments for the Author):

The study provides significant insights into the role of trimethylamine in rescuing morphological defects in *Streptomyces venezuelae*. The experiments are well-designed and the results are clearly presented. The manuscript is nearly ready for

publication, with only minor revisions needed to enhance clarity and organization (please refer to my comments).

Reviewer #2 (Comments for the Author):

The manuscript by Zhu et al. describes the conditional rescue of sporulation in the *S. venezuelae* MU-1 (bld or whi mutant) through dual culture with the wild-type strain. They revealed that TMA produced by the wild-type *S. venezuelae* was responsible for rescuing the sporulation of MU-1. Although the observed phenomenon is interesting, I feel that the presented data are preliminary and require additional evidence to meaningfully support the conclusion. Additionally, there are several technical concerns that should be addressed (as pointed out below). Therefore, I do not support publication in its current format.

Here I have several suggestions:

1. The authors discussed culture conditions that do not require glucose depletion for the production of TMA, which differs from previous observations. To focus on the culture conditions for producing TMA, the authors may need to systematically investigate the medium requirements for producing a sufficient amount of TMA that can rescue the phenotype of MU-1. Since the relatedness of medium contents was ambiguous among the four media that authors used, the authors may need to determine which component is important. For example, authors could use the N-Evans-CA agar medium and identify the components that impact TMA production by systematically removing each component.

2. If the focus is on the genetic background for the response to TMA by MU-1, I believe the authors should resequence the genome of the MU-1 mutant to identify point mutations. This would help determine which gene is responsible for the bld (or whi) phenotype. Although the authors describe it as a bld phenotype, from the SEM images, it appears that aerial mycelium is formed. Additionally, the authors might consider performing random mutagenesis and screening for mutants that do not sporulate in the presence of TMA. This approach could help identify the gene(s) required for the response to TMA in sporulation (morphological development).

These are the major concerns:

Please provide details on how MU-1 was generated?

L170-173:

How did the pH change over time (as shown in Fig. 2)? Were the levels of ammonium and TMA the same?

L215-216:

Please provide details on the function of the genes (*mtrA*, 5003, *orrA*) that give a bald phenotype.

L231, 242:

"sp." should not be italic.

please explain why the authors used different strains: *S. lividans* 1326 (L231) and *S. lividans* TK24 (L247)?

L312:

please provide details on how MU-1 was plated?

L314:

"The containers were refilled as needed to ensure that reagent was present."

> please specify the exact volume and timing.

L316-323:

Quantification of TMA

The method you used does not seem appropriate for the extraction of volatiles. Please present the UHPLC chromatogram and include the conditions used for the analysis (L323).

L316:

"Equal numbers of spores"

Please provide the exact number of spores, as the initial cell number (CFU or cell weight) at inoculation affects the growth of *Streptomyces*.

L317:

The authors described that MU-1 is deficient in sporulation. Please provide details on how to prepare the spores from MU-1.

Fig. 3B:

Is this divided by wet cell weight or dry cell weight? Please specify.

Additionally, why did the author extract TMA only from the cells? Why was the diffuse part in the media or vapor phase ignored? Please provide a specific reason.

Fig. 5B day 10:

The photo is not aligned to the center. Please show the whole plate (Similarly for all figures). It seems the part to be clipped is not unified.

Rescue of morphological defects in *Streptomyces venezuelae* by the alkaline volatile
compound trimethylamine

Yanping Zhu^a, Yanhong Zeng^a, Meng Liu^b, Ting Lu^a, and Xiuhua Pang^{a*}

6 ^aThe State Key Laboratory of Microbial Technology, Shandong University, Qingdao 266237, China.

7 ^bSchool of Municipal and Environmental Engineering, Shandong Jianzhu University, Jinan 250101,
China

*For correspondence, Email: pangxiuhua@sdu.edu.cn

Running title: Alkaline volatile promotes the growth of *Streptomyces*

Key words: trimethylamine, alkaline volatile, *Streptomyces*, morphology

[revised manuscript text omitted]

exposed to TMA, ammonium, or water by adding the liquids to a small cylindrical container
positioned in the center of each plate. The containers were refilled as needed to ensure that reagent
was present.

**Quantification of TMA.** Equal number spores of the wild-type strain *S. venezuelae* ISP5230
and the mutant strain MU-1 were inoculated onto N-Evans-CA medium (six regular petri dishes for
each strain) and were incubated at 30°C for ten days before the mycelium of each strain was
harvested. To extract the metabolites, 100 µl of water was first added to 100 mg of harvested
*Streptomyces* mycelium, followed by vortexing for 60 seconds, and then 400 µl of
methanol-acetonitrile (1:1, v/v) was added, followed by sonication (30 min) twice, and then the
mixture was centrifuged (14000rpm) for 20 min. The supernatant was used for
ultra-high-performance liquid chromatography (UHPLC) analysis.

**Scanning electron microscopy (SEM).** For SEM analysis, sterile coverslips were inserted into
N-Evans-CA medium inoculated with spores, followed by incubation for 96 h for *S. venezuelae*
ISP5230 or MU-1, and by incubation for 168 h for MU-1 co-incubated with ISP5230. Next, the
coverslips were transferred to a solution containing 2.5% glutaric acid for fixation, followed by
washing 2-3 times with PBS buffer (pH 7.0). Afterwards, the coverslips were treated with PBS
buffer containing 1% osmic acid for 2 h followed by another wash with PBS buffer. A gradient of
ethanol solution (30%, 50%, 70%, 80%, 90%, and 100%) was used for dehydrating the coverslips.
Finally, the coverslips were dried by a critical point dryer (LEICA EM CPD300), coated with metal
ions using ion sputtering and metal coating equipment (Cressinton Sputter Coater 108), and finally
viewed with a scanning electron microscope (FEI Quanta 250 FEG, USA) as described(17, 26).

**Author contributions** XP conceived and supervised the study. YZhu, YZeng, ML, and TL
performed the experiments. XP wrote the manuscript. All authors read and approved the
manuscript.

**Funding** This work was supported by grants from the National Natural Science Foundation of
China (NSF32270072 to XP), from the Open Funding Project of State Key Laboratory of Microbial

Metabolism (MMLKF24-04 to XP), and from the China Postdoctoral Science Foundation
(2023M732083 to YZhu).

**Declaration of competing interest**

The authors declare that they have no known competing financial interests or personal relationships
that could have appeared to influence the work reported in this paper.

**Reference**

- 1. **Kanchiswamy CN, Mainoy M, Maffei ME.** 2015. Chemical diversity of microbial volatiles and their
potential for plant growth and productivity. *Front Plant Sci* **6**.
- 2. **Netzker T, Shepherdson EMF, Zambri MP, Elliot MA.** 2020. Bacterial Volatile Compounds: Functions in
Communication, Cooperation, and Competition. *Annual Review of Microbiology*, Vol 74, 2020 **74**:409-+.
- 3. **Audrain B, Farag MA, Ryu CM, Ghigo JM.** 2015. Role of bacterial volatile compounds in bacterial biology.
*Fems Microbiology Reviews* **39**:222-233.
- 4. **Audrain B, Létoffé S, Ghigo JM.** 2015. Airborne Bacterial Interactions: Functions Out of Thin Air? *Frontiers*
*in microbiology* **6**.
- 5. **Misztal PK, Lymeropoulou DS, Adams RI, Scott RA, Lindow SE, Bruns T, Taylor JW, Uehling J,**
**Bonito G, Vilgalys R, Goldstein AH.** 2018. Emission Factors of Microbial Volatile Organic Compounds from
Environmental Bacteria and Fungi. *Environ Sci Technol* **52**:8272-8282.
- 6. **Hopwood DA (ed.).** 2007. *Streptomyces in Nature and Medicine.* OXFORD UNIVERSITY PRESS.
- 7. **Chater K.** 2011. Differentiation in *Streptomyces*: the properties and programming of diverse cell-types, p.
43-86. *In* P D (ed.), *Streptomyces: Molecular Biology and Biotechnology.* Caister Academic Press.
- 8. **Avalos M, Garbeva P, Raaijmakers JM, van Wezel GP.** 2020. Production of ammonia as a low-cost and
long-distance antibiotic strategy by *Streptomyces* species. *Isme J* **14**:569-583.
- 9. **Jones SE, Ho L, Rees CA, Hill JE, Nodwell JR, Elliot MA.** 2017. *Streptomyces* exploration is triggered by
fungal interactions and volatile signals. *Elife* **6**.
- 10. **Elliot MA, Buttner, M.J., and Nodwell, J.R.** 2008. Multicellular development in *Streptomyces*, p. 419-438.
*In* Whitworth DE (ed.), *Myxobacteria: Multicellularity and Differentiation* American Society for
Microbiology.
- 11. **Létoffé S, Audrain B, Bernier SP, Delepierre M, Ghigo JM.** 2014. Aerial Exposure to the Bacterial Volatile
Compound Trimethylamine Modifies Antibiotic Resistance of Physically Separated Bacteria by Raising
Culture Medium pH. *Mbio* **5**.
- 12. **Jones SE, Pham CA, Zambri MP, McKillip J, Carlson EE, Elliot MA.** 2019. Volatile Compounds Influence
Exploration and Microbial Community Dynamics by Altering Iron Availability. *Mbio* **10**.
- 13. **Briard B, Heddergott C, Latgé JP.** 2016. Volatile Compounds Emitted by *Pseudomonas aeruginosa*
Stimulate Growth of the Fungal Pathogen *Aspergillus fumigatus*. *Mbio* **7**.
- 14. **Kieser T, Bibb MJ, Buttner MJ, Chater KF, Hopwood DA (ed.).** 2000. *Practical Streptomyces Genetics*, 2nd
Edition ed. Norwich: John Innes Foundation
- 15. **Ou X, Zhang B, Zhang L, Zhao G, Ding X.** 2009. Characterization of *rrdA*, a TetR family protein gene
involved in the regulation of secondary metabolism in *Streptomyces coelicolor*. *Appl Environ Microbiol*
**75**:2158-2165.
- 16. **Bibb MJ, Domonkos A, Chandra G, Buttner MJ.** 2012. Expression of the chaplin and rodlin hydrophobic

- sheath proteins in *Streptomyces venezuelae* is controlled by sigma(BldN) and a cognate anti-sigma factor,
 RsbN. *Mol Microbiol* **84**:1033-1049.
- 17. **Zhang P, Wu L, Zhu Y, Liu M, Wang Y, Cao G, Chen XL, Tao M, Pang X.** 2017. Deletion of MtrA inhibits
 cellular development of *Streptomyces coelicolor* and alters expression of developmental regulatory genes.
 *Frontiers in microbiology* **8**:2013.
- 18. **Zhu Y, Wang X, Zhang J, Ni X, Zhang X, Tao M, Pang X.** 2022. The regulatory gene wblA is a target of the
 orphan response regulator OrrA in *Streptomyces coelicolor*. *Environmental Microbiology* **24**:3081-3096.
- 19. **Camarena-Pozos DA, Flores-Núñez VM, López MG, Partida-Martínez LP.** 2021. Fungal volatiles emitted
 by members of the microbiome of desert plants are diverse and capable of promoting plant growth.
 *Environmental Microbiology* **23**:2215-2229.
- 20. **Bentley SD, Chater KF, Cerdeno-Tarraga AM, Challis GL, Thomson NR, James KD, Harris DE, Quail
 MA, Kieser H, Harper D, Bateman A, Brown S, Chandra G, Chen CW, Collins M, Cronin A, Fraser A,
 Goble A, Hidalgo J, Hornsby T, Howarth S, Huang CH, Kieser T, Larke L, Murphy L, Oliver K, O'Neil
 S, Rabbinowitsch E, Rajandream MA, Rutherford K, Rutter S, Seeger K, Saunders D, Sharp S, Squares
 R, Squares S, Taylor K, Warren T, Wietzorrek A, Woodward J, Barrell BG, Parkhill J, Hopwood DA.**
 2002. Complete genome sequence of the model actinomycete *Streptomyces coelicolor* A3(2). *Nature*
 **417**:141-147.
- 21. **Ikeda H, Ishikawa J, Hanamoto A, Shinose M, Kikuchi H, Shiba T, Sakaki Y, Hattori M, Omura S.** 2003.
 Complete genome sequence and comparative analysis of the industrial microorganism *Streptomyces*
 *avermitilis*. *Nat Biotechnol* **21**:526-531.
- 22. **Gust B, Challis GL, Fowler K, Kieser T, Chater KF.** 2003. PCR-targeted *Streptomyces* gene replacement
 identifies a protein domain needed for biosynthesis of the sesquiterpene soil odor geosmin. *Proc Natl Acad Sci*
 *U S A* **100**:1541-1546.
- 23. **Garbeva P, Avalos M, Ulanova D, van Wezel GP, Dickschat JS.** 2023. Volatile sensation: The chemical
 ecology of the earthy odorant geosmin. *Environmental Microbiology* **25**:1565-1574.
- 24. **Becher PG, Verschut V, Bibb MJ, Bush MJ, Molnár BP, Barane E, Al-Bassam MM, Chandra G, Songs
 LJ, Challis GL, Buttner MJ, Flärdh K.** 2020. Developmentally regulated volatiles geosmin and
 2-methylisoborneol attract a soil arthropod to bacteria promoting spore dispersal. *Nat Microbiol* **5**:821-827.
- 25. **Fink D, Weissschuh N, Reuther J, Wohlleben W, Engels A.** 2002. Two transcriptional regulators GlnR and
 GlnR^{II} are involved in regulation of nitrogen metabolism in *Streptomyces coelicolor* A3(2). *Mol Microbiol*
 **46**:331-347.
- 26. **Liu M, Zhang P, Zhu Y, Lu T, Wang Y, Cao G, Shi M, Chen XL, Tao M, Pang X.** 2019. Novel
 two-component system MacRS is a pleiotropic regulator that controls multiple morphogenic membrane protein
 genes in *Streptomyces coelicolor*. *Appl Environ Microbiol* **85**:e02178-02118.

**Figure captions**

**Fig. 1.** The *S. venezuelae* mutant strain MU-1 has a defective morphology. (A) Phenotypes of the *S.*
 *venezuelae* wild-type strain ISP5230 (WT) and the mutant strain MU-1 grown at 30°C on solid
 N-Evans-CA medium (120 h) on separate plates. (B) SEM images of ISP5230 (WT) and MU-1 after
 growth on N-Evans-CA agar for 5 days, and MU-1 after 10 days of co-incubation with ISP5230 on
 N-Evans-CA agar (MU-1+volatiles). (C) Phenotypes of alternating patches of strain ISP5230 (WT)
 and MU-1 grown at 30°C on the same plate of solid N-Evans-CA medium (96 h).

**Fig. 2.** Co-incubation with *S. venezuelae* strain ISP5230 rescues the morphological defects of MU-1.
(A, B) Strain ISP5230 (WT) and MU-1 were grown at 30°C on solid N-Evans-CA medium on (A) a
standard petri dish without a physical barrier and (B) a special petri dish with a physical barrier
between the two strains. Images of the top of the plate were taken at the indicated times.

**Fig. 3.** The *S. venezuelae* WT strain ISP5230 produces TMA (trimethylamine). (A) The temporal
pH values of the *S. venezuelae* strain ISP5230 (WT) growth medium and the blank (uninoculated)
medium. (B) Quantification of TMA production. *S. venezuelae* strains ISP5230 (WT) and MU-1
were grown at 30°C on solid N-Evans-CA medium for 10 days. The mycelia were harvested, and
TMA or its oxide form (TMAO) was quantified by UHPLC. Data are the means with standard
deviations from six different preparations.

**Fig. 4.** The effects of exposure to ammonium (A), water (B), and TMA (C) on the growth of strain
MU-1 at 30°C on solid N-Evans-CA medium. Water (control) or the tested reagents were added to
the yellow container in the center of the plate, and plates were incubated for the indicated times.

**Fig. 5.** Evaluation of the rescue of the morphological defects of MU-1 by *S. venezuelae* strain
ISP5230 on different media. Co-incubation of *S. venezuelae* strain ISP5230 (WT) grown on YBP
(A), MS (B), and MYM (C) media with strain MU-1 grown on N-Evans-CA medium. The petri
dishes contained physical barriers between the two strains.

**Fig. 6.** Evaluation of the rescue of different strains with a bald phenotype by *S. venezuelae* strain
ISP5230. Strain ISP5230 (WT) and MU-1 were grown on solid N-Evans-CA medium for 6 days.
The two *S. venezuelae* mutant strains ($\Delta mtrA_{SVEN}$ and $\Delta 5003_{SVEN}$) and the two *S. coelicolor* mutant
strains ($\Delta mtrA_{SCO}$ and $\Delta orrA_{SCO}$) were grown on YBP medium and incubated with strain ISP5230 in
petri dishes with physical barriers for 10 days.

**Fig. 7.** Co-incubation with *S. venezuelae* strain ISP5230 promotes the growth of other *Streptomyces*
species. Strain ISP5230 (WT) were grown on MS and *S. lividans* TK24 were grown on YBP at
30°C on a special petri dish with a physical barrier between the two strains. Images of the top of the
plate were taken at the indicated times.

A

Day 2

Day 6

Day 10

WT

MU-1

**B**

WT

MU-1

**C**

WT

MU-1

WT

WT

WT

WT

WT

MU-1

$\Delta mtrA_{SVEN}$

$\Delta 5003_{SVEN}$

$\Delta mtrA_{SCO}$

$\Delta orrA_{SCO}$

Day 2

Day 4

Day 6

Day 8

Day 10

WT

TK24

TK24

TK24

We thank the Editor and Reviewers for their helpful suggestions, and we address the comments below. For additional clarity, we have also included a new file with figures for review only (see “supplementary materials for reviewers only”).

Reviewer #1:

Question 1, line 45, briefly mention the specific methods used for the key findings

Reply: The Abstract was revised to include the methods used for the key findings (lines 37-41), and additional details have been added in Methods and Materials (lines 330-355).

Question 2, line 48, statement about the novelty and significance of the research

Reply: The novelty and significance of our research has been clarified (lines 57-62).

Question 3, line 306, “physical barriers”, more details on these barriers

Reply: Information was added about the physical barrier as suggested (line 317). Furthermore, a picture of an empty petri dish with a barrier was added as ‘supplementary materials for reviewers’ (figure 1) for clarity.

Question 4, Ensure that all citations are correctly formatted and complete according to the journal's guidelines. References should be more up to date.

Reply: References were formatted as suggested, and more updated references were cited.

Reviewer #2:

Suggestion 1. The authors discussed culture conditions that do not require glucose depletion for the production of TMA, which differs from previous observations. To focus on the culture conditions for producing TMA, the authors may need to systematically investigate the medium requirements for producing a sufficient amount of TMA that can rescue the phenotype of MU-1. Since the relatedness of medium contents was ambiguous among the four media that authors used, the authors may need to determine which component is important. For example, authors could use the N-Evans-CA agar medium and identify the components that impact TMA production by systematically removing each component.

Reply: In this study, we had tried several routinely used growth media including MS, MYM, YBP, and N-EVANS for *Streptomyces* and found that the wild-type *S. venezuelae* can produce alkaline volatiles when grown on MS, YBP, and N-EVANS. We noticed that these three media contain glucose, in contrast to the original growth conditions used to produce TMA, which were reported in the pioneer work by Stephanie E Jones et al. (eLife 2017;6:e21738). Consequently, we thought that it was important to report that *S. venezuelae* can produce alkaline volatiles on multiple routinely used media. However, although investigating the medium requirement for

producing TMA is of interest, the main focus of our current study was on intraspecies rescue rather than on identifying the specific components that influence TMA levels.

Suggestion 2. If the focus is on the genetic background for the response to TMA by MU-1, I believe the authors should resequence the genome of the MU-1 mutant to identify point mutations. This would help determine which gene is responsible for the bld (or whi) phenotype. Although the authors describe it as a bld phenotype, from the SEM images, it appears that aerial mycelium is formed. Additionally, the authors might consider performing random mutagenesis and screening for mutants that do not sporulate in the presence of TMA. This approach could help identify the gene(s) required for the response to TMA in sporulation (morphological development).

Reply: The focus of this study is that we determined that alkaline volatiles, mainly TMA, produced by *S. venezuelae* under several growth conditions could rescue the growth defect of a mutant strain; although its genetic background is not clear yet, the mutant is certainly a derivative of *S. venezuelae*, and therefore TMA enables intra-species rescue. However, the genetic background of MU-1 is of interest to us, and these suggestions are very much appreciated. In fact, we are planning to conduct RNA-Seq analysis to determine differentially expressed genes between the wild-type strain and MU-1, with the goal of identifying potential genes that might contribute to TMA synthesis. We also plan to sequence the genome of MU-1, as suggested, to find mutations that may be associated with the reduced TMA production and the bld phenotype in this strain. In the long term, we may perform random mutagenesis and screen for mutants that do not sporulate in the presence of TMA to identify the associated gene(s), as suggested, although this is beyond the scope of our current study. Regarding the presence of some aerial mycelium in the mutant, we have revised the statement to indicate that aerial mycelium was minimal and that primarily vegetative mycelium was formed (lines 111-112).

Major concerns:

Question 1, Please provide details on how MU-1 was generated?

Reply: Information on the generation of MU-1 was provided as suggested (lines 98-100).

Minor concerns:

Question 1, L170-173: How did the pH change over time (as shown in Fig. 2)? Were the levels of ammonium and TMA the same?

Reply: The pH change over time was added as supplementary figure 4 with a description in lines 175-177. 1.0M of TMA and 1.2 M of ammonium were used in this assay. As now indicated, ammonium and TMA had similar effects on the pH.

Question 2, L215-216: Please provide details on the function of the genes (mtrA, 5003, orrA) that give a bald phenotype.

Reply: More information on the function of genes (mtrA, 5003, orrA) was added (lines 222-228).

Question 3, L231, 242: "sp." should not be italic.

Reply: Revised as suggested (lines 242, 252, 255).

Question 4, please explain why the authors used different strains: *S. lividans* 1326 (L231) and *S. lividans* TK24 (L247)?

Reply: In fact, there was no special reason for that. When we were conducting the co-incubation experiment with WT *S. venezuelae* and other species, we happened to have enough of the TK24 spores, and so we decided to include this strain. As it turned out, this strain was the only one among those species that responded to the alkaline volatiles produced by WT *S. venezuelae*.

Question 4, L312: please provide details on how MU-1 was plated?

Reply: Details were provided on how MU-1 was plated as suggested (lines 323-326).

Question 5, L314: "The containers were refilled as needed to ensure that reagent was present." > please specify the exact volume and timing.

Reply: Information about filling and refilling was provided as suggested (lines 323-329).

Question 6, L316-323: Quantification of TMA. The method you used does not seem appropriate for the extraction of volatiles. Please present the UHPLC chromatogram and include the conditions used for the analysis (L323).

Reply: Extraction of metabolites and subsequent analysis were performed by the Shanghai Applied Protein Technology Co., Ltd. Using the protocol supplied by the company, a detailed method was provided as suggested (lines 334-355). The UHPLC chromatogram for metabolite standards (the only one that the work report provided) was provided in figure 3 of the supplementary materials for reviewers.

Question 7, L316: "Equal numbers of spores" Please provide the exact number of spores, as the initial cell number (CFU or cell weight) at inoculation affects the growth of *Streptomyces*.

Reply: The exact number of spores was provided as suggested (line 331).

Question 8, L317: The authors described that MU-1 is deficient in sporulation. Please provide details on how to prepare the spores from MU-1.

Reply: MU-1 is deficient in sporulation on N-EVANS-AA medium. However, it produces spores when grown on other medium such as MS; therefore, we prepared the spores for MU-1 using cultures grown on MS medium. We have clarified this point on lines 330-331.

Question 9, Fig. 3B: Is this divided by wet cell weight or dry cell weight? Please specify.

Additionally, why did the author extract TMA only from the cells? Why was the

diffuse part in the media or vapor phase ignored? Please provide a specific reason.

Reply: In Fig. 3B, the value was divided by wet cell weight, which was specified as suggested (line 193). For metabolomics analysis, immediate fast sampling and freezing of the sample is preferred to preserve the cellular metabolites. Furthermore, the volatiles will be evaporated completely during the drying process (normally over 60 °C) if dry cells are used.

The metabolomics analysis was performed by the Shanghai Applied Protein Technology Co., Ltd. As requested by the company, we needed to supply wet cells of no less than 200 mg, and therefore, only wet cells were used in this study. In addition, the distribution of TMA and other metabolites might not be even in the diffused media, which could lead to misinterpretation and fluctuations for data analysis. Also, our culture systems were not suitable for capturing the vapor phase for this type of analysis. Nevertheless, our results clearly show distinct differences in TMA and TMAO production between the WT and MU-1

Question 10, Fig. 5B day 10: The photo is not aligned to the center. Please show the whole plate (Similarly for all figures). It seems the part to be clipped is not unified.

Reply: As suggested, a new Fig 5 was prepared using images taken from the same plate as suggested. Additionally, an original image showing the entire plate (prepared in duplicate) at 10 days of growth was included as figure 2 in supplementary materials for reviewers.

Re: Spectrum01195-24R1 (Rescue of morphological defects in *Streptomyces venezuelae* by the alkaline volatile compound trimethylamine)

Dear Dr. Xiuhua Pang:

Thank you for carefully addressing the concerns the reviewers had for your initial submission and also for providing additional image files. I only have one further comment to make: Could you please add the image of the split plate to show the barrier to the supplementary material too? It would be beneficial to all readers to have this image available, not only for the reviewers.

Your manuscript has been accepted, and I am forwarding it to the ASM production staff for publication. Your paper will first be checked to make sure all elements meet the technical requirements. ASM staff will contact you if anything needs to be revised before copyediting and production can begin. Otherwise, you will be notified when your proofs are ready to be viewed.

Sincerely,
Katharina Kujala
Editor
Microbiology Spectrum